# Socioeconomic Factors Related to Job Satisfaction among Formal Care Workers in Nursing Homes for Older Dependent Adults

**DOI:** 10.3390/ijerph18042152

**Published:** 2021-02-23

**Authors:** Isabel Pardo-Garcia, Roberto Martinez-Lacoba, Francisco Escribano-Sotos

**Affiliations:** 1Facultad de Ciencias Económicas y Empresariales, Universidad de Castilla-La Mancha, 02071 Albacete, Spain; isabel.pardo@uclm.es (I.P.-G.); francisco.esotos@uclm.es (F.E.-S.); 2Departamento de Economía Política, Hacienda Pública, Estadística Económica y Empresarial y Política Económica, Universidad de Castilla-La Mancha, 02071 Albacete, Spain; 3Centro de Estudios Sociosanitarios (CESS), Universidad de Castilla-La Mancha, 02071 Albacete, Spain; 4Grupo de Investigación en Economía, Alimentación y Sociedad (GEAS), Universidad de Castilla-La Mancha, 02071 Albacete, Spain; 5Departamento de Análisis Económico y Finanzas, Universidad de Castilla-La Mancha, 02071 Albacete, Spain

**Keywords:** socioeconomic factors, job satisfaction, long-term care workers, nursing homes

## Abstract

Population ageing is increasing the demand for dependent care. Aged care nursing homes are facilities that provide formal care for dependent older persons. Determining the level of job satisfaction among workers in nursing homes and the associated factors is key to enhancing their well-being and the quality of care. A cross-sectional survey was administered online to nursing home workers (*n* = 256) in an inland region of Spain over the period from February to May 2017. The questionnaire collected data on sociodemographic variables and others related to training and job satisfaction. The results show that most of the care is delivered by women with a medium level of education. A total of 68% of workers had received formal training, although a significant percentage (65%) thought this was not useful. The highest level of satisfaction was found to be related to users and co-workers. Our factor analysis revealed that the satisfaction components are decision-making, working conditions—e.g., schedule—and the work environment—e.g., relationship with coworkers—. Length of service and working with highly dependent persons are negatively associated with these components. Working in social health care is negatively related to decision-making and working conditions. Training, in contrast, is positively associated with these components. Care is a job that requires appropriate training and preparation to provide quality assistance and to guarantee workers’ well-being.

## 1. Introduction

Population ageing means the future of long-term care entails greater demand, greater spending, more workers, and above all, higher expectations that the final years of life should have the greatest possible meaning, purpose and well-being [1]. Ageing, then, gives rise to challenges and opportunities from a social, economic, political, and financial perspective. The demographic structure of Spain and Castilla-La Mancha show a country and an Autonomous Community with a percentage of population aged over 65 was around 20% in January 2019 [2], this percentage is forecast to increase to 30% or higher [3], which implies a future growth in demand for care.

The provision for long-term care varies considerably from one country to another, and even within a country itself. Alongside informal care, provided by family and friends, formal care is gaining prominence, especially in high-income countries. The Spanish Law 39/2006 of 14 December on the Promotion of Personal Autonomy and Care for Dependent Persons [4], of a universal nature, also laid out the set of formal services. Included in these services is care in nursing homes, that is, residential facilities providing services and intervention programmes tailored to the needs of such persons [5]. In Spain, there exist 372,985 places in residential care facilities, of which 27.16% are publicly administered and 72.84% are privately run [6], which covers 4.09% of the total population of persons aged over 65. In Castilla-La Mancha the number of available residential places is 26,649, with 37.49% being publicly owned facilities and 62.51% privately owned, representing a coverage of 6.90% of the population older than 65 years of age.

Broadly speaking, current discussion revolves around how to sustainably meet the needs of long-term care, but focuses less on the nature and quality of care and the support received [3]. These elements are, however, key to maintaining older people’s integrity, respect, and quality of life. In residential homes, different types of workers share the same working space: administrative staff, social and health care workers, and maintenance staff. These workers and administrators sometimes lack adequate knowledge, training, and support, which can lead them to feel dissatisfied with their work, potentially impacting, in turn, on the quality of care provided. Analysing job satisfaction in social and health care workers is important for health economics and healthcare finance, as well as organisational behaviour. Job satisfaction has an impact on productivity, performance, absenteeism, retention, recruitment, organizational commitment and patient satisfaction and care [7]. The traditional model of job satisfaction is grounded in an individual’s feelings towards their work, but what makes a job more, or less, satisfactory depends not only on the nature of the work, but also on an individual’s expectations of what they can bring to the job [8].

In this sense, literature has examined the influence of organisations on job satisfaction [9], causes of dissatisfaction among healthcare staff [10] or the necessity of training to provide skills and knowledge among facility administrators [11,12]. The situation of certain groups, such as social workers has also received attention [13,14,15]. In the specific field of long-term care, attention has focused on both nurses and care aides, but not in other workers. The job satisfaction of nurses in homes for older adults, as well as the quality of care and nurses’ health, has been analysed by Choi et al. [16], Schmidt et al. [17], and Westermann et al. [18]. Squires et al. [19] conducted a systematic review on non-professional carers in residential long-term care facilities, and Chamberlain et al. [20] carried out a study on care aide job satisfaction in Canada. In Spain, no nationwide periodic survey collects such data, but there are some studies related to this field. Ruiz-Fernández et al. [21] examine the quality of life and associated factors among nurses in the public health service, while another set of studies analyses the same elements in the field of long-term care. Fité-Serra et al. [22] focus on the working conditions of nursing staff in care homes. Sarabia-Cobo et al. [23] study the relationship between stress and quality of life in psychogeriatric professionals, and Briones-Peralta et al. [24] analyse the job satisfaction of carers in a nursing home for older persons with dementia.

To the best of our knowledge, studies have used small samples and have focused on specific group of workers. Despite our work has not a larger sample than in other studies (i.e., Choi et al. [16], Chamberlain et al. [20] or Ruiz-Fernández et al. [21]), it contributes to the previous literature in two gaps: first, we analyse the workers as a whole from a sample of nursing homes—not only focusing on specific groups as nurses or nursing assistant—; and second, we analyse their job satisfaction. Thus, the aim of the present work is to analyse job satisfaction among workers of different professional categories in residential care homes in order to determine the factors associated with satisfaction and so make recommendations to enhance such satisfaction.

## 2. Materials and Methods

### 2.1. Design and Participants

This study was conducted in the autonomous community of Castilla-La Mancha, Spain. An ad hoc cross-sectional questionnaire was administered in 10 residential nursing homes for older persons, two in each of the region’s five provinces—the total number of nursing homes in this autonomous community is 216. The research team explained the questionnaire to representatives of staff and managers in meetings. These meetings were done before the collection of the data, explaining the questionnaire, the objective of the study and the importance of the participation. The questionnaire was answered by 256 workers of different professional categories (see the Appendix A for the questionnaire): management, administration, health care, social care and maintenance. Each participant answered individually the questionnaire in printed form—anonymously—and then they sent the answer by postal mail. We obtained a response rate of 64% of the workers and they represented an 8% of the nursing homes workers of the region. The questionnaire was divided into three blocks across 39 items. The first block gathered data on sociodemographic variables, the second covered training-related aspects, while the third block contained questions about job satisfaction. Our fieldwork was carried out from February to May 2017.

### 2.2. Variables

The sociodemographic variables included gender, level of education (no completed studies, basic education, primary education, vocational training Grade 1, vocational training Grade 2, higher secondary, lower university degree, higher university degree and master’s degree/PhD), age, marital status (single, married/living together, divorced, widowed) and size of the municipality of residence (<500, 500–1000, 1000–3000, 3000–5000, 5000–10,000, 10,000–20,000 and >20,000 inhabitants). Respondents were also asked about the position held, type of working day (split shift, full working day, part-time, etc.), contractual status (temporary or permanent), rank (employee, supervisor, middle management and senior management), length of service in the company (<1 year, 1–3 years, 3–5 years, 5–10 years and >10 years), distance from home to place of work (<1 km, 1–5 km, 5–20 km, 20–50 km and >50 km), level of residents’ dependency (mild, moderate and severe), ownership of the institution/workplace (public, private, state subsidised) size of workplace municipality (<500, 500–1000, 1000–3000, 3000–5000, 5000–10,000, 10,000–20,000 and >20,000 inhabitants), service provided at work and teamwork (if they have done team work and they think that teamwork improves results). In addition, some of the categories of these variables were merged, as size of municipality of residence and workplace (<20,000, >20,000 inhabitants), distance from home to place of work (<5 km, >5 km), level of education (lower university degree to master’s/PhD), working day (one category for “As needs requires”, “Flexible” and “On call”) and rank (merging middle and senior management).

With regard to training, respondents were asked whether they received training specific to their position, whether the company provided continuous training, whether they received training related to their position in the company, the length of any training, whether they had attended talks, seminars, congresses or other similar activities. The section on level of job satisfaction included 17 items, scored from 1 (not at all satisfied) to 5 (very satisfied): schedule, workplace safety, workload, rest periods, information received from the company, working environment, relationship with co-workers, relationship with users, opportunities for promotion and/or improvement, supervision and/or coordination with superiors, relationship with superiors, support and motivation received from the company, autonomy in decision-making, possibility to take part in company decisions, possibility of negotiating working conditions, comfortable working environment and, finally whether they enjoyed their job.

### 2.3. Statistical Analysis

We first conducted a contingency analysis to explain the study variables. We then performed a factor analysis on the job satisfaction variables to determine the specific factors associated with the level of workers’ job satisfaction in the nursing homes included in the study. Finally, using a stepwise multiple regression analysis, we identified the personal and professional characteristics associated with each of the factors obtained. The variables included in this model, their treatment and name change (if done) are detailed in the Appendix A (Appendix A). All the statistical analyses were conducted using Stata 16 [25].

## 3. Results

Table 1 shows the descriptive analysis of the sociodemographic variables and those related to working conditions. Of the respondents, 80% were women. A total of 50% of the sample had completed either an intermediate level of vocational training or higher secondary education and 42% were aged between 41 and 55 years. The most common marital status was “married”, almost 54%, while 63% of respondents lived in municipalities with a population of more than 20,000 inhabitants.

In the nursing home, care aides and cleaning and catering staff accounted for more than 67% of the sample. More than 50% of the workers worked in shifts and nearly 77% had a permanent contract. For most workers, this was their only job, and they had no dependent persons in their charge (83.27%). Practically 44% of workers had more than 10 years of service. Over 60% of the workers lived less than 5 kilometres from their workplace. Of the nursing home residents, 53% presented a high degree of dependency. The employees generally worked in teams and view this as a way to achieve better results. As regards ownership of the nursing homes, 58% of the respondents worked in publicly owned facilities.

The block of questions on training shows, in Table 2, that 68% of the respondents has been given training and 57% of such training was delivered by the company where they work. A total of 27% of the workers had never been given training and the proportion of employees without training over the last two years is 35%. The reported training consists mainly of short courses, with 47% of these being of a duration of less than 10 h. The training is considered to be ineffective by 65% of the respondents.

The third block of questions covered variables related to the level of workers’ job satisfaction. Table 3 shows the results for each of the levels of satisfaction, as well as the mean and standard deviation for each of the satisfaction variables under study. In this case, the item with which workers exhibit the greatest satisfaction is their relationship with the users, followed by their relationship with co-workers. In contrast, the items with which they show the highest level of dissatisfaction are the possibility of deciding/decision-making and the possibility of negotiating their working conditions.

A factor analysis was conducted in order to group the satisfaction variables into a smaller number of components. This analysis showed the job satisfaction variables could be grouped together into three factors (Figure 1). The first component comprises the variables related to decision making–autonomy, possibility to decide and possibility of negotiation. The second includes the variables connected to satisfaction with working conditions—schedule, safety, workload and rest periods, and finally, the third component covers the variables related to the working environment—working environment, relationship with co-workers, relationship with users. The results reveal that the variance explained by these three factors is 65.69%. Appendix A and Appendix A in the Appendix A show the description of independent variables included in the multiple regression full model and the eigenvalues of the selected factors and the variance according to the varimax orthogonal rotation conducted.

In addition, Table 4 shows the results of the multiple regression analysis performed using the extracted factors as dependent variables with a confidence level of 90%. For the first component of satisfaction, decision-making, the results shows that the factors positively associated with this construct are the following: having received previous training, the facility being public, considering that teamwork improves results, the company providing training, considering training to be useful and the level of responsibility in the position occupied. Negatively associated we find length of service, working in the area of social health care and dealing with residents with severe dependency.

For the second component, working conditions, the positively associated factors are as follows: having received previous training, the level of responsibility in the position occupied, considering training to be useful and the company providing training. Negatively associated are length of service, dealing with residents with severe dependency and working in the area of social health care.

Finally, for the third component, working environment, the nursing home being publicly owned is positively associated with greater satisfaction, as are level of education, considering training to be useful and thinking that teamwork improves results. The negatively associated variables are length of service and dealing with users with severe dependency.

## 4. Discussion

This study analyses the satisfaction and working conditions of workers from different professional categories, all of whom are involved in providing services in aged care nursing homes in the Castilla-La Mancha region of Spain.

Most of the workers in the nursing homes in the sample are women. This coincides with previous studies reporting that, in both low- and high-income countries, the majority of carers are women, in both the formal and informal sectors [26]. A study conducted in Canada showed that care provided by sons delayed institutionalization but had negative effects on the sons’ work and personal life [27].

There is an unfounded belief that caregiving is easy but, in fact, providing care for older adults with or without dementia requires appropriate training. Many professional carers are unprepared for the demands of their work and lack suitable training [28]. In this sense, the level of education of our sample was medium—vocational training and higher secondary education. Of the workers, 57% reported having received training although 35% had received none over the last two years. In addition, 65% of workers say that the training is not relevant to their work, which does not accord with the results obtained in previous works where the importance of training and of providing workers with information and interventions so they can deliver care with less stress and greater satisfaction [26,29].

According to the World Health Organization, to ensure the supply of care, it is essential to improve the working conditions of care workers, be it by providing training, career opportunities, appropriate workloads, flexible working hours and giving carers autonomy in decision-making [3]. In this regard, our results show that the level of the care workers’ satisfaction with their schedule is high, but not so in the case of workload, nor in the possibility of negotiating working conditions, participating in decision-making in the company or autonomy to make decisions. Previous studies have shown that having autonomy and the opportunity to make decisions is positively related to levels of job satisfaction [30,31], which coincides with the first component of our model, decision-making. The third component, the work environment, includes relationships with co-workers and relationships with users. Interpersonal relationships and patient care are key elements of job satisfaction [7,32].

The present study has also addressed which variables are associated with each of the three components obtained for the model—decision-making, working conditions and work environment. We found that length of service and having patients with severe dependency were negatively associated with all three components. With regard to length of service, a study has also shown that the longer staff have worked in a nursing home, the lower is their level of job satisfaction [33], although others have shown that length of service is not significant [19,30]. The negative association with working with patients with severe dependency may partly be explained by the fact that nursing home workers are more vulnerable and have lower levels of job satisfaction, as well as stress and burnout due to their workload [18,23]. In addition, working in social and health care was negatively associated with two components: decision-making and working conditions. These findings partly coincide with the literature on organizational factors in long-term care homes that concludes facilities’ resources and employees’ workload affect job satisfaction [11,19]. Undoubtedly, these three variables should be studied and worked upon to ensure that the three components of job satisfaction identified in our study are improved and so reduce the risk of burnout, a problem that has a direct impact on workers and an indirect impact on care and quality of service.

Importantly, many of the study variables related to training are positively associated with the three components—the company provides training, usefulness of training and having received previous training. This may be because training, whether received previously or in-service helps raise awareness of the work to be done and fosters synergies among workers. Indeed, although the reviews conducted by Dilig et al. [30] on nurses in health care systems, and by Squires et al. [19] in nursing homes, suggest that special training is not key, other studies underline the importance of providing specialised training in coping with the care of persons with severe dependency to enhance the working environment and prevent burnout [24,34,35,36,37].

Public ownership of the care facility is positively associated with two of the three components—decision-making and working environment. This may be because public institutions allow greater participation in decision-making processes and the relationships between co-workers and with residents. In addition, the working environment might be more sympathetic due to differences in their management systems, more horizontal vs. more vertical organisation, for example. It has been shown that care aides in privately run nursing homes suffer greater job insecurity [22].

Supporting care workers at work and in life and valuing them for what they do is important and enhances job satisfaction [1]. Despite this representing a cost for institutions, it should be considered as investing in the well-being of workers, and by extension, that of users. Feeling part of a team enhances job satisfaction, which is important in a setting where care labour markets need to promote policies that help retain workers [38]. Training people that are part of formal long-term care systems is key to improve their satisfaction and also to provide high-quality care, enhancing the well-being of both carers and care recipients.

### Limitations

This study has certain limitations, and the results should be interpreted with caution. Firstly, there might be a response bias in the questionnaire—e.g., workers indicating they are more/less satisfied with an item than they really are. Secondly, the study uses a cross-sectional survey and, thus, has the limitations inherent in this type of methodology —e.g., usually, cross-sectional studies show association/correlation and not causality—. Thirdly, the sample is limited to one autonomous community, or region, and so caution is needed in extrapolating/generalizing the findings. Fourth, 37.49% of facilities are public owned, but 58% of survey respondents work in publicly owned facilities. It could be explained because: (1) selected public nursing homes are larger than the private ones; and for this reason, (2) there are more workers in public nursing homes; (3) public managers could have a greater compromise with research. Fifth, we used an ad hoc questionnaire which means that it was specifically elaborated to develop this study.

## 5. Conclusions

Analysing job satisfaction in social and health care workers is essential for health economics and healthcare finance, as well as organisational behaviour. Job satisfaction has an impact on productivity, performance, retention, among others. Among long-term care workers, the highest level of satisfaction is related to users and co-workers. The satisfaction components are decision-making, working conditions —e.g., schedule— and the work environment —e.g., relationship with coworkers—. Training is positively associated with these components. Our results are relevant to policymakers. Due to the future need of long-term workers, a better understanding of the components of job satisfaction in nursing homes could help to develop better recruitment and retention policies in care sector. And, in addition, if they want to improve quality assistance and to guarantee workers’ wellbeing, training and preparation are key.

## Figures and Tables

**Figure 1 ijerph-18-02152-f001:**
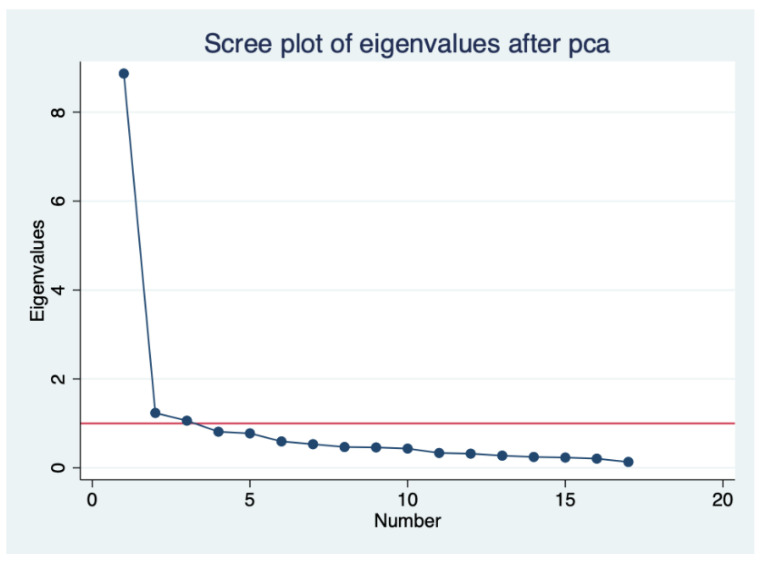
Scree plot of eigenvalues.

**Table 1 ijerph-18-02152-t001:** Sociodemographic characteristics and working conditions.

Sex	*N* (%)	Working Day	*N* (%)
Female	205 (80.1)	Shifts	126 (50.2)
Male	51 (19.9)	Full day	73 (29.1)
**Level of Education**		Split shift	22 (8.8)
No completed studies	5 (2.0)	Part–time	21 (8.4)
Basic education	13 (5.2)	As needs require/Flexible/On Call	9 (3.6)
Primary Education	30 (11.9)	**Type of Contract**	
Vocational training Grade 1 and Grade 2	100 (39.7)	Temporary	60 (23.4)
Higher secondary	28 (11.1)	Permanent	196 (76.6)
Lower university degree/Higher university degree/Master’s degree/PhD	76 (30.2)	**Rank**	
**Age**		Employee	209 (83.3)
16–25	20 (7.9)	Supervisor	26 (10.4)
26–40	86 (34.1)	Middle and senior management	16 (6.4)
41–55	106 (42.1)	**Length of Service**	
>55	40 (15.9)	<1 year	32 (12.7)
**Marital status**		1–3 years	44 (17.4)
Single	87 (34.7)	3–5 years	28 (11.1)
Married/Living together	135 (53.8)	5–10 years	38 (15.0)
Divorced	22 (8.8)	>10 years	111 (43.9)
Widowed	7 (2.8)	**Distance from Home to Workplace**	
**Size municipality of residence**		<5 km	156 (61.7)
<20,000	92 (36.7)	>5 km	97 (38.3)
>20,000	159 (63.3)	**Users’ Level of Dependency**	
**Size municipality of workplace**		Mild	48 (18.9)
<20,000	61 (25.8)	Moderate	71 (28.0)
>20,000	175 (74.2)	Severe	135 (53.2)
**Position**		**Teamwork**	
Nursing home duties	84 (33.5)	No	19 (7.5)
Psychology/Social work	6 (2.4)	Yes	233 (92.5)
Care aides	86 (34.3)	**Teamwork Improves Results**	
Health care	37 (14.7)	No	18 (7.1)
Maintenance	17 (6.8)	Yes	236 (92.9)
Management/Administration	18 (7.2)		
Leisure	3 (1.2)		

**Table 2 ijerph-18-02152-t002:** Training received by workers.

Has Received Previous Training	*N* (%)
No	80 (31.5)
Yes	174 (68.5)
The Company Provides Training	
No	109 (42.9)
Yes	145 (57.1)
Has Received Training	
Never	68 (26.7)
In the last 2 years	90 (35.3)
In the last year	28 (11.0)
In the last 6 months	56 (22.0)
In the last month	13 (5.1)
Length of Training	
<10 h	88 (47.1)
10–20 h	37 (19.8)
20–50 h	47 (25.1)
>50 h	15 (8.0)
Usefulness of Training	
No	165 (65.0)
Yes	89 (35.0)

**Table 3 ijerph-18-02152-t003:** Level of worker satisfaction.

Level of Satisfaction	*N* (%)					Mean (SD)
	1	2	3	4	5	
Schedule	28 (11.0)	30 (11.8)	74 (29.1)	56 (22.1)	66 (26.0)	3.4 (1.3)
Workplace safety	27 (10.7)	30 (11.9)	75 (29.6)	76 (30.0)	45 (17.8)	3.3 (1.2)
Workload	60 (23.5)	55 (21.6)	67 (26.3)	47 (18.4)	26 (10.2)	2.7 (1.2)
Rest periods	43 (16.9)	34 (13.4)	68 (26.8)	56 (22.1)	53 (20.9)	3.2 (1.3)
Information provided by the company	51 (20.1)	54 (21.3)	65 (25.6)	47 (18.5)	37 (14.6)	2.9 (1.3)
Working environment	43 (17.0)	41 (16.2)	68 (26.9)	73 (28.9)	28 (11.1)	3.0 (1.2)
Relationship with co-workers	15 (5.9)	15 (5.9)	57 (22.4)	88 (34.5)	80 (31.4)	3.8 (1.1)
Relationship with users	3 (1.2)	3 (1.2)	44 (17.3)	95 (37.4)	109 (42.9)	4.2 (0.9)
Promotion opportunities	71 (28.0)	50 (19.7)	70 (27.6)	42 (16.5)	21 (8.3)	2.6 (1.3)
Supervision and coordination with superiors	37 (14.5)	43 (16.9)	49 (19.2)	72 (28.2)	54 (21.2)	3.2 (1.4)
Relationship with superiors	26 (10.2)	35 (13.8)	55 (21.7)	74 (29.1)	64 (25.2)	3.4 (1.3)
Motivation from company	56 (22.0)	48 (18.8)	68 (26.7)	52 (20.4)	31 (12.2)	2.8 (1.3)
Autonomy in decision-making	50 (19.7)	43 (16.9)	78 (30.7)	52 (20.5)	31 (12.2)	2.8 (1.3)
Possibility to take part in company decisions	85 (33.7)	57 (22.6)	65 (25.8)	33 (13.1)	12 (4.8)	2.3 (1.2)
Possibility of negotiation working conditions	84 (33.2)	55 (21.7)	61 (24.1)	36 (14.2)	17 (6.7)	2.4 (1.2)
Comfortable working environment	35 (13.8)	39 (15.4)	77 (30.3)	70 (27.6)	33 (13.0)	3.1 (1.2)
Doing a job they enjoy	22 (8.7)	34 (13.5)	61 (24.2)	64 (25.4)	71 (28.2)	3.5 (1.3)

**Table 4 ijerph-18-02152-t004:** Multivariate regression analysis of the components obtained.

Component 1. Decision-Making			
**Independent Variables**	**Beta Coefficient**	**Standard Error**	***p***
Has received previous training	0.775	0.335	0.022
Publicly owned or not	0.740	0.322	0.023
Teamwork improves results	1.043	0.545	0.057
Length of service	−0.562	0.101	<0.001
Company provides training	1.386	0.315	<0.001
Work in social and health care or not	−0.710	0.360	0.050
Severe dependency or not	−0.744	0.293	0.012
Usefulness of training	0.663	0.308	0.032
Responsibility in position	0.562	0.246	0.023
**Component 2. Working Conditions**			
**Independent Variables**	**Beta Coefficient**	**Standard Error**	***p***
Length of service	−0.339	0.074	<0.001
Severe dependency or not	−0.511	0.229	0.027
Has received previous training	0.878	0.262	0.001
Responsibility in position	0.396	0.193	0.041
Usefulness of training	0.426	0.242	0.079
Position	−0.868	0.271	0.002
Company provides training	0.660	0.242	0.007
**Component 3. Working Environment**			
**Independent Variables**	**Beta Coefficient**	**Standard Error**	***p***
Publicly owned or not	0.622	0.196	0.002
Level of education	0.086	0.050	0.083
Severe dependency or not	−0.356	0.177	0.045
Length of service	−0.338	0.066	<0.001
Usefulness of training	0.523	0.191	0.007
Has received previous training	0.388	0.201	0.055
Teamwork improves results	0.644	0.344	0.062

## Data Availability

Data available on request.

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
