# Peer review of "Socioeconomic Factors Related to Job Satisfaction among Formal Care Workers in Nursing Homes for Older Dependent Adults"

_ijerph, 2021, doi:10.3390/ijerph18042152_

Round 1

Reviewer 1 Report

Thank you for the privilege to read you paper. I agree with what you wrote in your introduction, that increasingly we will need to attract people to these important jobs and retain them. This study will help to that end.

Abstract: It is concise and informative. You may want to use words that differentiate and more specifically describe “work conditions” versus “work environment”.

Paper’s conclusion – (same)

Introduction:  It is long, reads like a review article, and meanders a bit. It is packed with information. I would consider more strictly clustering the themes and eliminated excessive verbiage and edited out anything tangential, as this distracts from the narrative of your research. Your narrative would be clearer and more impactful if you could condense the intro into three short segments – background on the subject you are investigating, gap in the literature, the aims of your study. All of the component parts are present but scattered throughout.

            Make sure your investigation stands out as unique, to fill a gap in the literature. You do this, and you cite previous investigations that have already been done on the subject. You may wish to clarify this.

Methods:  I questioned who was included in the study. While I understand that you want to make sure that everyone needed to run a home for the ageing be included in the study, however your endpoints are related to direct hand-on care (ex those severely dependent on care). I would not expect management/administration and maintenance would not have hands-on experience like this. They may not receive the same training. And management/administration likely has a lot more autonomy in decision-making. Also, the management/administration and maintenance experience may not be specific to homes for the elderly, and if you were to ask them about their stressors it may be less likely to be related to direct care of the severely dependent patients, and more likely due to rule and regulation, reimbursement, and administrative complexity.  (I am not sure what the “leisure” position does.)  I think your results would be more meaningful if you only included those that provide direct care for the elderly residents. If you decide to include everyone, I think you have to make the case for it in your methods.

Discussion:  The comment about men carers p. 9 lines 220-223 seems tangential, awkward, perhaps an opinion. Do you need it to tell the story?

P9, line 229, did you mean to use the word “contrast”?

Limitations: I think you need to say why there may be a response bias, why there are limitations to a cross-sectional survey.

Conclusions: It seems from your introduction that you did this study because you saw there would be a need for more care-givers for the ageing population and you wanted to know what would make the job more appealing or find high yield opportunities to reduce stressors. Your conclusion is your opportunity to make your recommendations to policymakers, I would look at your word choices and make sure that the conclusion would be clear to your audience if it is the only paragraph they read. (Similar to your abstract.)

Thank you for the privilege to review the first draft of your manuscript. I think your introduction is correct in describing how we will need to draw more people into these important jobs and retain them. I hope your work will help with that effort.

Author Response

Thank you for the privilege to read you paper. I agree with what you wrote in your introduction, that increasingly we will need to attract people to these important jobs and retain them. This study will help to that end.

Authors: Thank you very much for your comments. They have undoubtedly made it possible to improve the work considerably.

Abstract: It is concise and informative. You may want to use words that differentiate and more specifically describe “work conditions” versus “work environment”.

Authors: Thank you for your comment. We have included an example for each one in the Abstract.

Paper’s conclusion – (same)

 Authors: OK. We have included the same examples than in the Abstract.

Introduction:  It is long, reads like a review article, and meanders a bit. It is packed with information. I would consider more strictly clustering the themes and eliminated excessive verbiage and edited out anything tangential, as this distracts from the narrative of your research. Your narrative would be clearer and more impactful if you could condense the intro into three short segments – background on the subject you are investigating, gap in the literature, the aims of your study. All of the component parts are present but scattered throughout.

Authors: Thank you. We have reorganized and synthesize the Introduction section following your recommendations.

Make sure your investigation stands out as unique, to fill a gap in the literature. You do this, and you cite previous investigations that have already been done on the subject. You may wish to clarify this.

Authors: OK. We have added a sentence at the end of the Introduction section indicating the gap we try to fill.

Methods:  I questioned who was included in the study. While I understand that you want to make sure that everyone needed to run a home for the ageing be included in the study, however your endpoints are related to direct hand-on care (ex those severely dependent on care). I would not expect management/administration and maintenance would not have hands-on experience like this. They may not receive the same training. And management/administration likely has a lot more autonomy in decision-making. Also, the management/administration and maintenance experience may not be specific to homes for the elderly, and if you were to ask them about their stressors it may be less likely to be related to direct care of the severely dependent patients, and more likely due to rule and regulation, reimbursement, and administrative complexity.  (I am not sure what the “leisure” position does.)  I think your results would be more meaningful if you only included those that provide direct care for the elderly residents. If you decide to include everyone, I think you have to make the case for it in your methods.

Authors: Thank you for your comment. As you have indicated, our intention in this paper is to analyze the satisfaction of everybody who is working in a nursing home as a whole. The main reason of that is that we want to obtain global results of all selected nursing homes —controlling variables despite the job position— to report managers and decision makers (e.g., politicians) problems or issues related to whole institutions.

You are right indicating that management and maintenance workers have not received the same training. However, they should receive specific courses related with caring —at least theoretically— because it could help to better understanding how the care workers and users could feel and then to generate synergies, better practices, and a better work environment.

On the other hand, leisure job position is people that work in the entertaining of the users, for example, workers that organize all leisure activities (games, travels, parties, etc.).

Discussion:  The comment about men carers p. 9 lines 220-223 seems tangential, awkward, perhaps an opinion. Do you need it to tell the story?

Authors: You are right, we have deleted the comment.

P9, line 229, did you mean to use the word “contrast”?

Authors:  Thanks for your comment. We have modified the text and now you can read: “In addition, 65% of workers say that the training is not relevant to their work, which does not accord with the results obtained in previous works where the importance of training and of providing workers with information and interventions so they can deliver care with less stress and greater satisfaction [29,32]”

Limitations: I think you need to say why there may be a response bias, why there are limitations to a cross-sectional survey.

Authors: OK. We have included an example of response bias and a limitation of a cross-sectional survey in Limitations subsection.

 Conclusions: It seems from your introduction that you did this study because you saw there would be a need for more care-givers for the ageing population and you wanted to know what would make the job more appealing or find high yield opportunities to reduce stressors. Your conclusion is your opportunity to make your recommendations to policymakers, I would look at your word choices and make sure that the conclusion would be clear to your audience if it is the only paragraph they read. (Similar to your abstract.)

Authors: Thank you very much for your comment. We have changed the Conclusions section and you can read: “Analysing job satisfaction in social and health care workers is essential for health economics and healthcare finance, as well as organisational behaviour. Job satisfaction has an impact on productivity, performance, retention, among others. Among long-term care workers, the highest level of satisfaction is related to users and co-workers. The satisfaction components are decision-making, working conditions and the work environment. Training is positively associated with these components. Our results are relevant to policymakers. Due to the future need of long-term workers, a better understanding of the components of job satisfaction in nursing homes could help to develop better recruitment and retention policies in care sector. And, in addition, because if they want to improve quality assistance and to guarantee workers’ wellbeing, training and preparation are key.”

Thank you for the privilege to review the first draft of your manuscript. I think your introduction is correct in describing how we will need to draw more people into these important jobs and retain them. I hope your work will help with that effort.

Authors: We thank the reviewer for making positive and constructive comments

Reviewer 2 Report

This paper analyzes sociodemographic data from eldercare workers of different professional categories with a focus on job satisfaction.  Job satisfaction is critical not only for the workers but also for the quality of care provided to the patients as well as the success of the operation; therefore, identifying correlations between the various factors can offer insights to the industry.  This is a valuable contribution to the existing literature.

Line 111 - This journal is published in English therefore the Annex should be translated into English, as well as provided in Spanish, as it was given.  It is important that the Spanish version of the survey remain, so please do not replace the actual survey given in Spanish with a translated English version.  Both Spanish (as given) and English (translated) should be provided in the Annex.

Line 148 - In the nursing "hose" should be "home"

Line 154 - From line 59, we learn that 37.49% of facilities are publicly owned, yet from line 154 we learn that 58% of survey respondents work in publicly owned facilities.  Perhaps the publicly owned facilities are significantly larger and employ many more people than the privately owned facilities.  Else, the study might be biased with a larger population of responding workers coming from publicly owned facilities than would otherwise be representative of the population of workers.  Further discussion and clarification of this is worthwhile to either declare bias or elaborate on how this % of respondents aligns to the private/public worker ratio in Castilla-La Mancha.  Later in the paper at lines 280-281, a reference to potential bias is noted, and this point may be what is being alluded to.  The paper would be strengthened by further elaborating on this topic.  

Author Response

Reviewer 2.

This paper analyzes sociodemographic data from eldercare workers of different professional categories with a focus on job satisfaction.  Job satisfaction is critical not only for the workers but also for the quality of care provided to the patients as well as the success of the operation; therefore, identifying correlations between the various factors can offer insights to the industry.  This is a valuable contribution to the existing literature.

Authors:

Thank you very much for your comments. They have help us to improve our work considerably.

Line 111 - This journal is published in English therefore the Annex should be translated into English, as well as provided in Spanish, as it was given.  It is important that the Spanish version of the survey remain, so please do not replace the actual survey given in Spanish with a translated English version.  Both Spanish (as given) and English (translated) should be provided in the Annex.

Authors: We have included a translated version of the questionnaire in the Annex.

Line 148 - In the nursing "hose" should be "home"

Authors: Thank you, you are right. We have changed “hose” for home.

Line 154 - From line 59, we learn that 37.49% of facilities are publicly owned, yet from line 154 we learn that 58% of survey respondents work in publicly owned facilities.  Perhaps the publicly owned facilities are significantly larger and employ many more people than the privately owned facilities.  Else, the study might be biased with a larger population of responding workers coming from publicly owned facilities than would otherwise be representative of the population of workers.  Further discussion and clarification of this is worthwhile to either declare bias or elaborate on how this % of respondents aligns to the private/public worker ratio in Castilla-La Mancha.  Later in the paper at lines 280-281, a reference to potential bias is noted, and this point may be what is being alluded to.  The paper would be strengthened by further elaborating on this topic.  

Authors: Thank you for your comment. When we design the study, we first tried to choose a proportional sample of nursing home depending on the ownership. But we could not control the response of the workers. It could be explained because: 1) selected public nursing homes are bigger than the private ones; and for this reason, 2) there are more workers in public nursing homes; 3) public managers could have a greater compromise with research.  We have included this explanation in Limitations subsection.

Regarding the potential bias, we have written an example for each one.